# Target Trial Emulation Using Hospital-Based Observational Data: Demonstration and Application in COVID-19

**DOI:** 10.3390/life13030777

**Published:** 2023-03-13

**Authors:** Oksana Martinuka, Maja von Cube, Derek Hazard, Hamid Reza Marateb, Marjan Mansourian, Ramin Sami, Mohammad Reza Hajian, Sara Ebrahimi, Martin Wolkewitz

**Affiliations:** 1Institute of Medical Biometry and Statistics, Faculty of Medicine and Medical Centre, University of Freiburg, 79104 Freiburg, Germany; 2Biomedical Engineering Department, Engineering Faculty, University of Isfahan, Isfahan 81746-73441, Iran; 3Biomedical Engineering Research Centre (CREB), Automatic Control Department (ESAII), Universitat Politècnica de Catalunya-Barcelona Tech (UPC) Building H, Floor 4, Av. Diagonal 647, 08028 Barcelona, Spain; 4Department of Epidemiology and Biostatistics, School of Health, Isfahan University of Medical Sciences, Isfahan 81746-73461, Iran; 5Department of Internal Medicine, School of Medicine, Isfahan University of Medical Sciences, Isfahan 81746-73461, Iran; 6Alzahra Research Institute, Alzahra University Hospital, Isfahan University of Medical Sciences, Isfahan 81746-75731, Iran

**Keywords:** competing events, COVID-19, methodology, observational data, target trial emulation

## Abstract

Methodological biases are common in observational studies evaluating treatment effectiveness. The objective of this study is to emulate a target trial in a competing risks setting using hospital-based observational data. We extend established methodology accounting for immortal time bias and time-fixed confounding biases to a setting where no survival information beyond hospital discharge is available: a condition common to coronavirus disease 2019 (COVID-19) research data. This exemplary study includes a cohort of 618 hospitalized patients with COVID-19. We describe methodological opportunities and challenges that cannot be overcome applying traditional statistical methods. We demonstrate the practical implementation of this trial emulation approach via clone–censor–weight techniques. We undertake a competing risk analysis, reporting the cause-specific cumulative hazards and cumulative incidence probabilities. Our analysis demonstrates that a target trial emulation framework can be extended to account for competing risks in COVID-19 hospital studies. In our analysis, we avoid immortal time bias, time-fixed confounding bias, and competing risks bias simultaneously. Choosing the length of the grace period is justified from a clinical perspective and has an important advantage in ensuring reliable results. This extended trial emulation with the competing risk analysis enables an unbiased estimation of treatment effects, along with the ability to interpret the effectiveness of treatment on all clinically important outcomes.

## 1. Introduction

The coronavirus disease 2019 (COVID-19) pandemic, caused by severe acute respiratory syndrome coronavirus 2, triggered unprecedented speed in generating evidence on treatment effectiveness through the use of real-world data [1,2,3]. However, estimating treatment effects from observational data is challenging and demands careful consideration [4]. Due to the lack of randomised treatment exposure assignment in an observational data context, two main limitations should be considered. First, treatment indication differences due to imbalanced prognostic characteristics between treatment groups can lead to baseline confounding bias [5,6]. Second, varied treatment initiation times can lead to immortal time bias [4,7]. Furthermore, competing risks need to be considered in survival data analysis. By definition, a competing risk is an event that prevents the occurrence of the primary outcome of interest [8]. Failure to account for one or several methodological biases yields inaccurate results and flawed conclusions [3,4,6,9]. For example, misclassification of immortal time, that is, when patients are classified as treated, from the study beginning leads to a survival advantage for the exposed group [4,10]. Ignorance of competing events and application of the naïve Kaplan–Meier method often results in overestimating the risk of a primary event [11,12,13].

Methodological biases are common in observational studies when assessing the effectiveness of treatment in hospitalised patients with COVID-19 [3,6]. A review of COVID-19 studies published in high-impact journals showed that immortal time, confounding, and competing risk biases were only partially controlled for or entirely ignored [3]. In particular, only 1 out of 11 evaluated studies addressed competing risk bias [3]. In-hospital mortality or the so-called ‘undesirable’ or ‘negative’ outcome is often considered as a primary event of interest in COVID-19 studies. In these studies, hospital discharge prevents the observation of in-hospital death. Similarly, in research investigating discharge alive or the so called ‘desirable’ or ‘positive’ outcomes, in-hospital death events prevent observing discharge alive. Therefore, competing events must be properly taken into account in time-to-event analyses involving hospital mortality or recovery-based end points to avoid flawed results [9,14,15,16]. However, naïve methods for survival analysis ignore competing risks.

During the COVID-19 pandemic, observational data from electronic records has been increasingly used to emulate target trials. This design framework can be applied to mimic a hypothetical trial or contribute to existing knowledge, especially when data from randomised controlled trials are unavailable [17]. During the COVID-19 pandemic, this framework was used in a wide range of evaluations, i.e., drug therapies [18,19,20,21,22,23,24], extracorporeal membrane oxygenation [25,26], vaccine effectiveness [27,28], non-pharmaceutical interventions, and policy evaluations [29]. To emulate a trial, there are two common methods: the cloning approach and the sequential trial approach [17,30]. The cloning approach involves creating two duplicates of each patient and distributing them to the two treatment arms, while the sequential trial approach mimics a sequence of time zeros [31]. Both approaches address potential immortal time bias but also require that weighting methods be adapted to account for selection and confounding biases [30,31]. This framework can also be extended in the presence of competing risks.

In this paper, we aim to emulate a trial and conduct an extended competing risk analysis. We present the methodology in the context of implementing the framework on real-world data from patients with COVID-19 hospitalized at the Medical Center Isfahan, Iran, during the pandemic’s first wave in 2020. By using a common example of data in our case study, we aim to describe the advantages and limitations of longitudinal observational patient data. For demonstration, we chose to evaluate an antiviral treatment’s effectiveness. To avoid drawing clinical conclusions from our methodological study, we chose not to show the pharmaceutical name of the evaluated treatment. In our analysis, we make a simplifying assumption by setting the hazard rates to time-constant. This translates into a closed and simplified version of the cumulative incidence function. The reason for this assumption is to deepen understanding of the challenge of competing risks. Furthermore, this assumption can be easily relaxed, as our study demonstrates.

The paper is organised as follows. In Section 2, we begin with a description of our emulated trial and specify its key components. The trial is then emulated using observational data via the three-step process of cloning, censoring, and weighting. We introduce a competing risks model and its associated statistical methods to extend the emulated trial. In Section 3, we present the results of the data example and compare them with sensitivity analyses and naïve traditional survival methods. In Section 4, we discuss the approach we proposed and compare it with other methods reported in the literature. Our model’s limitations are described in Section 5. Finally, our summary and conclusions are found in Section 6.

## 2. Materials and Methods

### 2.1. Emulated Trial Specification

To emulate a target trial, a detailed protocol must be drafted that describes its underlying design and analytic components [17]. In our study, the emulated trial protocol included study questions, outcomes, eligibility criteria, treatment strategies and assignment, follow-up times, a grace period, an estimand, and a statistical analysis plan. A summary of our emulated trial’s key components is found in Table 1. All patients gave their written informed consent to the treatment procedure.

Our main clinical research question was to evaluate the effect of the antiviral treatment on the primary outcome, that is, in-hospital mortality among patients hospitalized with COVID-19. We also aimed to examine differences in hospital-stay durations between the two treatment groups. Discharge alive was considered as a competing event because it prevents observing in-hospital death between the time of discharge and the end of follow-up. All hospitalized patients experienced one of these two outcomes. The length of hospital stay was then measured as the time from hospital admission to either in-hospital death or discharge alive.

Patients were eligible (i) if they were 18 years of age or older, (ii) had laboratory-confirmed severe acute respiratory syndrome coronavirus 2 infection detected by polymerase chain reaction or had radiological evidence using high-resolution chest computed tomography showing lesions compatible with COVID-19, and (iii) were hospitalized from February to April, 2020 [32]. Patients were diagnosed according to the Chinese COVID-19 diagnosis and treatment guidelines as well as World Health Organization provisional advice [33]. Inclusion and exclusion criteria of our study’s target population are in Table 1, and a flowchart of patients eligible for a trial emulation is shown in Appendix A.

We evaluated the effectiveness of ‘X’ antiviral treatment with the standard of care versus the standard of care alone, in accordance with the treatment strategies used in clinical practice. We excluded the name of the evaluated treatment to avoid a clinical interpretation of our methodological study. Two strategies were compared, namely, initiating ‘X’ treatment within the first two days after hospital admission (in the text referred to as ‘treated’) versus no ‘X’ treatment during the first two days after admission, which meant no antiviral treatment at all, or antiviral treatment initiation after the first two days (‘untreated’). The decision on assigning ‘X’ treatment was based on provisional treatment recommendations [33]. The treatment was indicated when the oxygen saturation fell below 95% and a loading dose of the drug was given twice daily for five days.

To emulate the target trial, we needed to define the start of follow-up, the so-called baseline or time zero. In our study, baseline was defined as the time of hospital admission. The end of follow-up was in-hospital death, discharge alive, or the end of follow-up on day 60, whichever occurred first. A grace period designated when treatment could be initiated after baseline. Implementing a grace period makes the treatment strategies used in clinical practice more pragmatic [17,34]. The length of the grace period should rely on a clinical decision and serve to reduce heterogeneity between patients [35]. Our study defined the grace period during which patients were eligible to begin treatment as two days after hospital admission. Consequently, patient’s ‘cloned’ data could appear in both treatment strategies within two days [36].

In our hypothetical trial protocol, we also determined that the estimated effect is the observational analogue of the intention-to-treat effect. Firstly, we defined that ‘X’ treatment could be initiated within the first two days of hospital admission. We then summarized information from the hospital records on the assigned treatment within the grace period. Furthermore, we did not capture deviations from the protocol beyond the grace period, similar to the intention-to-treat principle. The simplified strategy we applied required ‘X’ treatment initiation or its non-initiation within the grace period, regardless of whether the treatment was continued. Patients who had been assigned ‘X’ treatment after the first two days were considered to be non-X-treated. Our main estimand of interest was risk differences for in-hospital death and discharge alive.

### 2.2. Practical Implementation of Cloning, Censoring, and Weighting

Emulating a target trial takes a three-step approach, namely cloning, censoring, and the inverse probability of artificial censoring weighting [31]. Cloning attempts mimic random allocation in that patients are assigned to both treatment strategies until their treatment strategy is confirmed [17,31,37]. In our case, each patient’s data were cloned at hospital admission and assigned to both the ‘X-treated’ arm and the ‘non-X-treated’ arm independently of their later treatment status, thus increasing the initial sample size. The trial arms were therefore identical with respect to the patients’ demographics and clinical characteristics at baseline. The cloning approach also enabled us to distribute patients who experience early fatal and nonfatal outcomes to both treatment strategy arms (Figure 1).

The emulated trial’s design implements artificial censoring by censoring patients when they deviate from the planned protocol during the grace period [37]. In our study, this meant that the patient given ‘X’ treatment within two days was censored at the treatment administration time point in the ‘non-X-treated’ arm. Similarly, patients in the ‘X-treated’ arm who did not start treatment within two days were censored at day two. Moreover, the person-time of patients who experienced in-hospital death or discharge alive outcomes within the grace period contributed to both arms (Figure 1).

The potential selection bias induced by artificial censoring can be corrected using the inverse probability of artificial censoring weights. The goal of the weights is to up-weight the remaining patients in the risk set so that they represent the censored patients. This step is important to maintain the comparability and balance of the two arms throughout the grace period [31,37]. To estimate the probability of being uncensored, we took a non-parametric approach. First, two Cox regression models were fitted separately for each treatment arm to assess the covariates’ (in linear form) influence on the probability of censoring mechanism. According to clinical knowledge, prognostic predictors, such as age, sex, oxygen saturation, respiratory rate, and creatinine serum level have influenced how physicians have assigned treatments. All covariates included in our model were considered by relying on the information available at baseline. Missing data on covariate values were not imputed, and records with missing data were excluded. Second, we estimated the probabilities of each patient remaining uncensored at day two of the grace period. Third, subject-specific unstabilized weights were calculated, inversely proportional to the estimated probability of remaining uncensored until the end of the grace period. Finally, the weights were incorporated in the outcome regression model [38]. Note that censored observations contributed to the denominator of the estimated constant hazards with the number of days in the original state (i.e., from the study entry until artificial censoring).

### 2.3. Statistical Analysis of the Emulated Trial

To evaluate the treatment effect on time-to-event outcomes, we took a three-step approach: (i) extending a composite end point survival analysis to a competing risks model; (ii) modelling a weighted cause-specific hazards regression for each outcome; (iii) incorporating cause-specific hazard rates to estimate cumulative incidence functions in order to obtain parameters enabling an interpretation of the results on risk rather than on rates. Strengthening the reporting of observational studies in epidemiology (STROBE) recommendations and a checklist are presented in Appendix A. All analyses described in this article were conducted using statistical software R (version 3.6.3). We used the *survival, boot*, and *ggplot2* packages in R.

#### 2.3.1. Competing Risk Framework

By definition, a competing risk (e.g., hospital discharge) is an event that prevents the occurrence of the primary outcome (e.g., in-hospital death) of interest [39]. Our emulated framework fits a simplified three-state competing risks model to estimate transition hazard rates (Figure 2), where in-hospital death and discharge alive are two absorbing events of the same disease [40,41]. Each arrow represents a time-homogeneous hazard rate $\lambda_{\mathrm{ij}}$ from the initial state of hospital admission as untreated or treated $\left(i=0,\;1\right)$ to the two terminating end points of in-hospital death $\left(j=2\right)\;$or discharged alive $\left(j=3\right)$, respectively. Hazards are interpreted as the instantaneous risk of moving from state i to state j [42]. To obtain a weighted version (Section 2.2), we incorporated the weights in the outcome models. In the main analysis, we considered a parametric model and relied on a time-homogeneous approach. We then applied our estimated constant hazard rates to calculate cumulative incidences and average lengths of hospital stay.

#### 2.3.2. Outcome Model: Cause-Specific Hazard Regression Model

The cause-specific model measures the association between a treatment exposure on the end point of interest (i.e., in-hospital death), and other events preventing the main outcome (i.e., discharged alive) are considered censored observations [43]. In a simple setting with time-constant hazards, a pooled logistic regression model can be applied to estimate the effect in the outcome model. The hazard ratios (HRs) are then approximated from the pooled logistic regression model due to the theoretical relationship between the odds ratio and HR. Specifically, two weighted pooled logistic regression models for the in-hospital death and discharged-alive outcomes including adjustment for the month of admission were fitted for each treatment arm separately according to the treatment status defined by the end of the grace period. The treatment effect was measured by contrasting transition-specific hazard rates, for example, λ_02_ versus λ_12_ for in-hospital death (Table 2).

#### 2.3.3. Outcome Model: Cumulative Incidence Function

In the presence of competing risks, each event’s probabilities are described in terms of cumulative incidence functions [8]. In the competing risk model (Figure 2), the estimated overall risk of moving from the initial state to state two at the end of follow-up (τ= 60 days) is related to the cause-specific rate. As shown, cumulative incidence functions can be obtained directly when the hazard rates for each transition are known (Table 2). We can estimate the absolute risk of experiencing an event by time *t* by relying on the cause-specific cumulative incidence function by considering the individuals who experienced competing events [40]. Assuming that hazards are time constant, cause-specific cumulative probabilities for in-hospital death (1) and discharged alive (2) in the untreated group are estimated as follows:(1)$$CIF_{02}\left(t\right)=\frac{\lambda_{02}}{\lambda_{02}+\lambda_{03}}\times \left(1-\exp\left(-\left(\lambda_{02}+\lambda_{03}\right)\times t\right)\right)$$
(2)$$CIF_{03}\left(t\right)=\frac{\lambda_{03}}{\lambda_{02}+\lambda_{03}}\times \left(1-\exp\left(-\left(\lambda_{02}+\lambda_{03}\right)\times t\right)\right)$$
where t is the time since hospital admission. 

In the final step, we evaluated cumulative incidences by comparing the cumulative risk of each outcome between treated and untreated patients. We also calculated the risk differences and risk ratios to quantify the treatment effect on the risk scale (Table 2). In the result section below, we relied on the 30-day cut-off point to visualize our results, while final estimates were obtained according to the 60-day follow-up period. This cut-off point was chosen because there were no significant differences in the results after 30 days. In addition, 95% confidence intervals were calculated by bootstrapping with 500 replicates. The code for the trial emulation was adapted according to the Maringe et al. tutorial [31]. The extended competing risk analysis statistical code is available upon request from the corresponding author. The data on this exemplary study are sensitive and not accessible to the public.

#### 2.3.4. Additional and Naïve Statistical Analyses

To evaluate our result’s robustness we conducted non-parametric estimation of the cumulative probabilities using the Aalen-Johansen approach. This estimator is used to study more complex transition probabilities, and it generalizes the Kaplan-Meier estimator in a competing risk setting. A detailed description and mathematical background of the non-parametric Nelson-Aalen and the Aalen-Johansen approach is found elsewhere [44]. We conducted additional analyses to assess the influence of the grace period’s choice, with one- and three-day grace periods. To demonstrate the potential impact of competing risks bias, we performed a naïve survival analysis using the standard Kaplan-Meier estimator. The Kaplan-Meier analysis (one minus the survival function) was used to calculate the cumulative probability of in-hospital death. The analysis was conducted using the original dataset (n = 618) without cloning.

## 3. Results

Among 655 hospitalized patients, 618 (94.4%) patients with complete information on the prognostic covariates were included in our original cohort. Baseline characteristics of the included patients are available in Appendix A. Following the protocol, a total of 618 patients were cloned in the treated arm (338 patients were categorized as X-treated and 280 censored), and 618 patients were cloned in the untreated arm (280 were categorized as non-X-treated and 338 censored) according to the two-day grace period (Appendix A). Of treated patients, 318 (94.1%) received ‘X’ treatment within the first day of their admission. Baseline patient characteristics were well-balanced between treatment groups (Appendix A).

### 3.1. Cause-Specific Cumulative Hazards, Cumulative Incidence Functions, and Risk Differences Taking the Constant Hazards Approach

During the 60-day follow-up period, there were 40 in-hospital deaths, whereas 578 patients were discharged alive. We detected no significant differences in the risk of hospital death or discharge between treated and untreated patients in our sample: HR: 0.79 (95% confidence interval (CI): 0.34 to 1.18) and HR: 1.03 (95% CI: 0.91 to 1.14), respectively (Appendix A). The cumulative in-hospital mortality rates were low during the 30-day cut-off period (Figure 3).

Our two treatment group’s estimated 60-day mortality risks were similar, for treated 5.8% and for untreated patients 7.4% (Appendix A). Since there was no right censoring, the corresponding discharge probabilities of each treatment arm were one minus the respective probability of in-hospital death (Appendix A).

The risk differences were clinically negligible between treatment arms over time (Figure 4).

The 60-day risk difference for in-hospital death was −0.016 (95% CI: −0.048 to 0.015) and 0.016 (95% CI: −0.015 to 0.048) for discharged alive outcomes (Appendix A). The estimated risk ratio for in-hospital death was 0.79 (95% CI: 0.34 to 1.17) and 1.02 (95% CI: 0.98 to 1.05) for discharged alive (Appendix A). The mean duration of hospital stay was 8 days in both treatment groups (Appendix A and Appendix A). ‘X’ treatment did not prove to be significantly associated with a shorter length of hospital stay at the 5% level. All summarised results of the corresponding measures presented in Table 2 are provided in Appendix A.

### 3.2. Additional and Naïve Analyses

In the additional analyses, the weighted cause-specific cumulative hazards obtained from the Nelson–Aalen estimator and the weighted cause-specific cumulative incidences using the Aalen–Johansen estimator revealed similar patterns (Appendix A). Subsequently, the risk differences were similar to the differences obtained from the parametric model (Appendix A). Our results were also compared with the grace period’s different choices, that is, the one- and three-day grace periods, which revealed no significant differences (Appendix A). In the naïve survival analysis using the Kaplan–Meier method, the cumulative probability of in-hospital death at 30-days was overestimated but did not significantly differ between treated (49.9%) and untreated (44.7%) patients (*p* > 0.05, differences: 7.1%) (Appendix A).

## 4. Discussion

In our study, the trial emulation with a competing risks model is proposed and applied to a setting where hospital data are available by adapting the step-by-step tutorial presented by Maringe et al. [31]. Taking the cloning approach allowed us to address the limitations associated with observational data, that is, immortal time bias and time-fixed confounding bias. The weighted version of the analysis enabled us to address the selection bias introduced by artificial censoring. Extending the framework to a competing risks model allowed us to account for competing risks and gain additional information on the effects of treatment.

The importance of recognising competing events in both randomised trials and observational studies has been emphasized previously [8,9,45]. Our study highlights how essential it is to also account for competing risks in emulated target trials. In studies relying on routine clinical data, the follow-up of COVID-19 patients often ends with either in-hospital death or discharge alive. In our study, we assumed that hospital discharge would serve as a valid proxy for recovering from the disease. This assumption is based on hospital readmissions and post-discharge mortality rates in COVID-19 patients being relatively low [46]. Using a demonstrative data example, we estimated the treatment effects on in-hospital death by considering the effects on the competing event of hospital discharge rather than eliminating it. At the analysis stage, we provided cause-specific hazards and cumulative event probabilities according to previous recommendations [8,47]. We must admit that discharge might not present a methodological issue in COVID-19 studies in which survival data beyond hospital discharge is available. If such information was available in our data, our statistical methods could have been simplified because discharge would no longer have been a competing event for death. In that case, for example, discharged patients remain in the risk set until a pre-specified time (e.g., date of last follow-up) [21].

Lin et al. compared three different analytical approaches to estimate survival probabilities in hospitalised patients with COVID-19 who received convalescent plasma therapy using observational data. They confirmed that the naïve (unweighted) Kaplan–Meier method yields biased results [48]. Similarly, in our naïve analysis, we showed that censoring patients at the time of the competing event leads to an overestimation of cumulative incidence probabilities for the primary outcome of both treatment groups. Therefore, applying standard survival analysis methods in the presence of competing risks is inappropriate [6,9,45,49]. Lin et al. also described two alternative approaches to account for competing events. Their first method assumed that the patients who were discharged would be alive until a pre-specified date. Their second method entailed an inverse probability weighted Kaplan–Meier estimator that accounted for the eliminated (censored) competing event of discharge alive [48]. We believe that acknowledging the treatment effects on discharge alive is more meaningful: the interest may be to examine not only how treatment prevents death, but also how treatment affects the probability of recovering from the infection [41]. We showed that a competing risks analysis provides important insights on treatment effects on all clinically meaningful and heterogeneous end points [11,50].

In conclusion, we suggest that different approaches could be considered to estimate treatment effects in the presence of competing events. This choice naturally depends on the research question of the investigation. The choice of approach may also depend on whether the outcome of interest is undesirable (e.g., in-hospital death) or desirable (e.g., discharge alive, recovery) [15]. The article by Young et al. provides detailed guidance on defining competing events and on analytical approaches, as well as on the required assumptions [51]. Finally, to estimate treatment effects consistent with those from randomised controlled trials, rigorous methodology is necessary. Thus, the contribution of observational data to the effectiveness of treatment evaluations using the trial emulated framework is only feasible when the target trial is appropriately emulated [36,52].

## 5. Limitations

Our study has important limitations. First, our main analysis relied on a straightforward estimation of time-constant hazards enabling basic understanding of the data, the concept of a competing risks approach, and statistical quantities [32]. However, the assumption of time-constant hazards limited our knowledge of fluctuations in the treatment effect during the observational period. Second, time-varying confounding remained underestimated due to the lack of information on time-updated prognostic covariates. This is despite our attempt to reduce the potential of time-varying confounding bias by choosing a grace period of two days and by considering as untreated those patients who initiated treatment later after the grace period. Furthermore, we assumed that our baseline confounders remained fixed within the grace period. Third, our simplified strategy was to compare initiators versus non-initiators, regardless of whether or not patients adhered to the treatment strategy to which they had been assigned after the grace period. This implied the estimation of an intention-to-treat effect. Evaluating dynamic treatment exposures and per-protocol effect estimation requires information on adherence, data availability on time-updated covariates, and the application of complex analytical approaches for time-varying confounders, such as g-estimation-based methods [53,54]. Due to these shortcomings, we were unable to interpret treatment effects as per-protocol effects [35].

## 6. Conclusions

We emulated this trial via an extended competing risk analysis applicable for evaluating the effectiveness of a treatment for hospitalised patients with COVID-19 by relying on routine register-based observational data. Our findings demonstrate that this methodology can overcome three major methodological challenges: immortal time bias, confounding bias, and competing risks. Our approach enables potential treatment effects to be evaluated on all clinically relevant outcomes and heterogeneous end points. We recommend avoiding application of the naïve Kaplan–Meier survival analysis method in the presence of competing events, as that can lead to exaggerated estimates. This paper and its supporting materials provide technical guidance for trial emulations involving competing risks analysis.

## Figures and Tables

**Figure 1 life-13-00777-f001:**
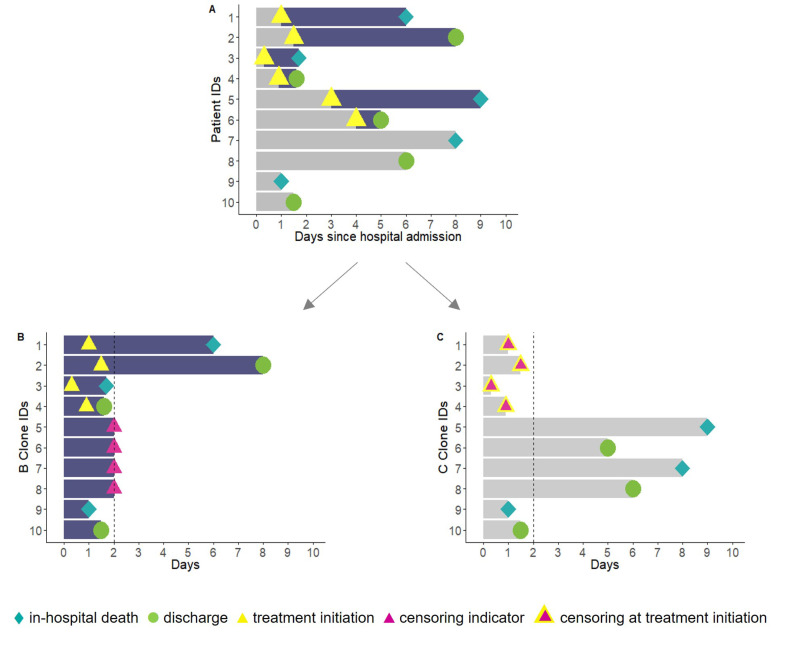
A hypothetical example of target trial emulation applying cloning and censoring techniques. Notes: In observed data (**A**), patients with identification (ID) numbers 1 to 6 received treatment at some time after hospital admission. Patients with ID 7-10 were never treated during their hospital admission. Patients with ID 1, 3, 5, 7, and 9 died, and 2, 4, 6, 8, and 10 were discharged alive. Two clones of the observed data were created for each patient, with one clone assigned to each treatment strategy. (**B**) All patients were assigned to the X-treated arm in cloned sample B, and patients were censored when they deviated from the planned protocol. Patients with IDs 5 to 8 who were never treated were censored at the end of the grace period. (**C**) All patients were assigned to the non-X-treated arm in cloned sample C, and patients were censored when they deviated from the planned protocol. Patients with IDs 1 to 4 treated within the grace period were censored at the time they received treatment.

**Figure 2 life-13-00777-f002:**
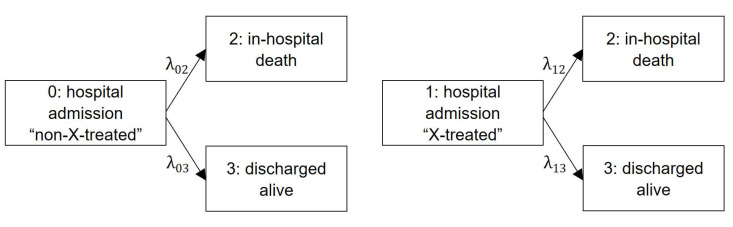
A competing risk model for trial emulation. Notes: Competing risk model with counterfactual (assuming all patients are treated or untreated from hospital admission) cause-specific hazards for in-hospital death and discharged alive, assuming that hazards are time-constant.

**Figure 3 life-13-00777-f003:**
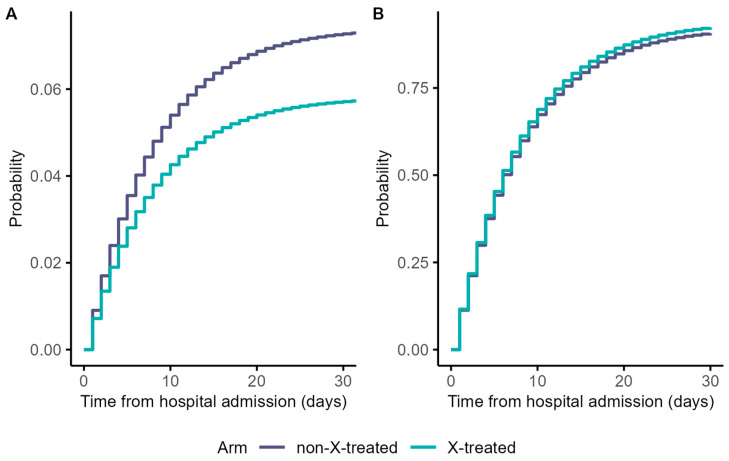
Weighted cumulative incidence function curves for in-hospital death (**A**) and discharged alive (**B**) outcomes. Notes: follow-up time of 60 days was shortened to 30 days for visualisation purposes.

**Figure 4 life-13-00777-f004:**
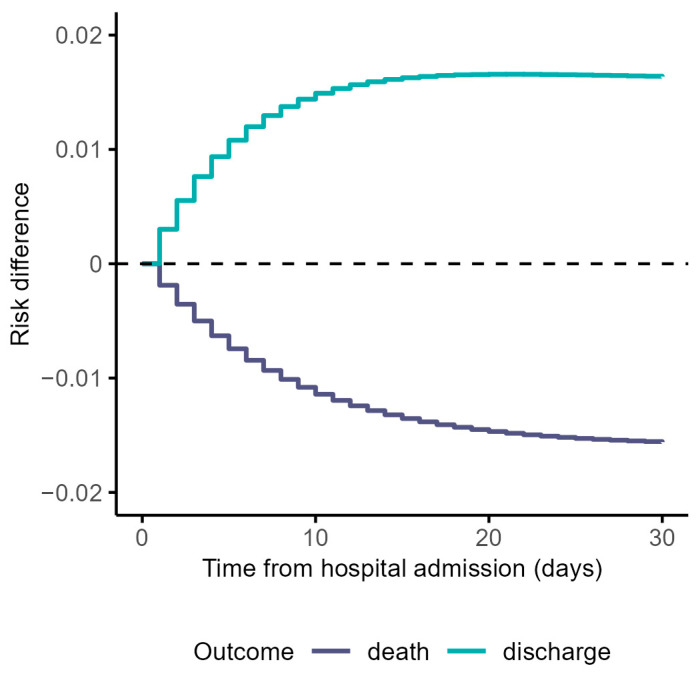
Risk differences estimated via weighted cumulative incidence functions for in-hospital death and discharged alive outcomes. Notes: follow-up time of 60 days was shortened to 30 days for visualisation purposes

**Table 1 life-13-00777-t001:** A summary of protocol components for the target trial emulation.

Protocol Component	Description of Emulation
**Research questions**	Is ‘X’ antiviral treatment associated with lower in-hospital mortality among patients hospitalized with COVID-19? Does the length of in-hospital stay differ between X-treated and non-X-treated patients?
**Outcomes**	In-hospital death and discharge alive (competing event)
**Eligibility criteria**	Adult patients aged ≥18 years PCR confirmed SARS-CoV-2 infection or HRCT chest findings Hospitalized within time period from February until May 2020
**Exclusions**	Any contraindication to ‘X’ antiviral treatment (e.g., liver dysfunction, kidney injury, cardiac arrhythmias, including QT prolongation) at hospital admission that made the patient unsuitable for receiving ‘X’ treatment
**Treatment strategies**	Initiate ‘X’ treatment (‘treated’) within two days of hospital admission Do not initiate ‘X’ treatment (‘untreated’) within two days of hospital admission
**Treatment assignment**	Non-randomized ‘X’ treatment assignment
**Follow-up time**	Begins with hospital admission, and treatment initiation must occur within the first two days after hospitalization and end at 60 days or in-hospital death or discharged alive
**Grace period**	First two days after hospital admission
**Estimand**	Difference in the risk for in-hospital death and discharge alive
**Analysis plan**	● Emulation of target trial: Analysis conducted on the cloned data set assuming random treatment assignment and with a censoring indicator when there is a protocol deviation within the grace period; selection bias accounted for by using inverse probability of artificial censoring weights. ● Type of primary analysis and statistical model: o Cause-specific hazard approach is selected to account for competing risks o Cause-specific discrete time-constant hazards for in-hospital death and discharge alive estimated via weighted pooled logistic regression o Cause-specific cumulative incidences estimated via weighted cumulative incidence functions o Differences in risk for in-hospital death and discharge alive
**Adjustment variables**	Demographic: age and sex Clinical characteristics: oxygen saturation, respiratory rate, and creatinine serum level measured at hospital admission

Abbreviations: COVID-19, Coronavirus disease 2019; HRCT, high-resolution computed tomography; PCR, polymerase chain reaction.

**Table 2 life-13-00777-t002:** Overview of weighted treatment effect measures using the parametric model assuming constant hazards.

Corresponding Measure ^a^	Mathematical Formulation
**Constant hazards**	
Death hazard *w*/*o* treatment	*λ* _02_
Discharge hazard *w*/*o* treatment	*λ* _03_
Hazard *w*/*o* treatment	$\lambda_{0}=\lambda_{02}+\lambda_{03}$
Death hazard with treatment	*λ* _12_
Discharge hazard with treatment	*λ* _13_
Hazard with treatment	$\lambda_{1}=\lambda_{12}+\lambda_{13}$
**Mortality**	
Mortality risk *w*/*o* treatment at the end of follow-up	$MR_{0}=\frac{\lambda_{02}}{\lambda_{02}+\lambda_{03}}$
Mortality risk with treatment at the end of follow-up	$MR_{1}=\frac{\lambda_{12}}{\lambda_{12}+\lambda_{13}}$
Mortality risk ratio at the end of follow-up	$\frac{\lambda_{12}/\lambda_{02}}{\left(\lambda_{13}+\lambda_{12})/(\lambda_{02}+\lambda_{03}\right)}$
Difference in mortality at the end of follow-up	$MR_{1}-MR_{0}$
**Hazards and cumulative incidence functions**	
Hazard ratio of death (treatment vs. *w*/*o* treatment) at the end of follow-up	$HR_{2}=\frac{\lambda_{12}}{\lambda_{02}}$
Hazard ratio of discharge (treatment vs. *w*/*o* treatment) at the end of follow-up	$HR_{3}=\frac{\lambda_{13}}{\lambda_{03}}$
Cumulative risk of death *w*/*o* treatment at time t	$CIF_{02}\left(t\right)=\frac{\lambda_{02}}{\lambda_{0}}\times \left(1-\exp\left(-\lambda_{0}t\right)\right)$
Cumulative risk of discharge *w*/*o* treatment at time t	$CIF_{03}\left(t\right)=\frac{\lambda_{03}}{\lambda_{0}}\times \left(1-\exp\left(-\lambda_{0}t\right)\right)$
Cumulative risk of death with treatment at time t	$CIF_{12}\left(t\right)=\frac{\lambda_{12}}{\lambda_{1}}\times \left(1-\exp\left(-\lambda_{1}t\right)\right)$
Cumulative risk of discharge with treatment at time t	$CIF_{12}\left(t\right)=\frac{\lambda_{13}}{\lambda_{1}}\times \left(1-\exp\left(-\lambda_{1}t\right)\right)$
**Risk differences and ratios**	
Risk difference functions for death at time t	$RD_{2}\left(t\right)=CIF_{12}\left(t\right)-CIF_{02}\left(t\right)$
Risk difference functions for discharge at time t	$RD_{3}\left(t\right)=CIF_{13}\left(t\right)-CIF_{03}\left(t\right)$
Risk ratios for death at time t	$RR_{2}\left(t\right)=\frac{CIF_{12}\left(t\right)}{CIF_{02}\left(t\right)}$
Risk ratios for discharge at time t	$RR_{3}\left(t\right)=\frac{CIF_{13}\left(t\right)}{CIF_{03}\left(t\right)}$
**Length of stay**	
Length of stay *w*/*o* treatment	$LOS_{0}=\frac{1}{\lambda_{02}+\lambda_{03}}$
Length of stay with treatment	$LOS_{1}=\frac{1}{\lambda_{12}+\lambda_{13}}$
Difference in length of stay	$LOS_{1}-LOS_{0}$

^a^ Inverse probability censoring weighted. Abbreviations: *CIF*, cumulative incidence function; *HR*, hazard ratio; *LOS*, length of stay; *MR*, mortality risk, *RD*, risk difference; *RR*, risk ratio; *w*/*o*, without. Notes: $\lambda_{0}$ = non-X-treated; $\lambda_{1}$ = X-treated.

## Data Availability

The code for the trial emulation was adapted according to the Maringe et al. tutorial [31]. The extended competing risk analysis statistical code is available upon request from the corresponding author. The data for this exemplary study are considered sensitive and not made publicly available.

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
