# Peer review of "Target Trial Emulation Using Hospital-Based Observational Data: Demonstration and Application in COVID-19"

_life, 2023, doi:10.3390/life13030777_

Round 1
Reviewer 1 Report
life-2208626-peer-review-v1
Target Trial Emulation Using Hospital-Based Observational Data: Demonstration and Application in COVID-19.
The author shares a study on applications of target trial emulation using hospital-based observational data in COVID-19. The topic itself is interesting. However, the article is poorly written, including issues in structuring, referencing, visualization, and many others.
Author should clarify the following:
1. Page 1, lines 18-27, please remove “Simple Summary”. It cannot be tolerated.
2. Page 2, the structure of the introduction section is not good. It should have two separate paragraphs at its end, one of which presents the contribution and explanations of this work; and the other one outlines the coming sections.
3. There is no smooth transition between the sections. It is not clear the relationship of sections 2 and 3 with 4. Please merge sections 3 & 4 to a single section renaming as “Results and Discussion”.
4. In abstract, some sentences are lengthy, and are written in past tense. The entire abstract should be in present tense. The author is suggested to revise all lengthy sentences available in abstract as well as in entire paper.
5. The manuscript is poorly structured. Why is a literature review jam-packed within the introduction and not presented as a standalone section?
6. It is not clear what theories and techniques are originally proposed by the authors. It seems only adopting existing ones. Please highlight novelty and innovation of this work.
7. Page 12, the CONCLUSION section can be improved. Authors should avoid marginal explanations. They need to focus on contribution and then on their achievements. It is also better to write the numerical values of the improvements. Verbs must also be in the past tense.
8. The information in the Introduction section is much too general, presenting facts that are already extremely well known about the pandemic (Covid-19), and not enough information regarding hydrogen oxygen therapy, which is the main focus of the study.
9. Page 2, Introduction is too poor. Its English very poor and I cannot understand it well. Also, the authors should mention the similar studies. For example, studies which have examined the risk of Covid-19 in the different geographical areas. Then, they should mention the Chinese studies and finally talk with regard to the knowledge gap that they would like to address. Furthermore, they should also mention the aim of the study at the end of the introduction part.
10. In addition, in introduction, before starting the mentioned references, there is a need to add 8-9 lines related to the subject of the paper and write in general introduction. After that you should connect them with the references.
11. The authors should use reference form all of the world. Covid-19 is an international problem and there are many similar studies in the world.
12. I am curious that are there any limitations to the proposed algorithm? If yes, the authors should identify it in the paper so that it would be an idea to improve the algorithm for future works.
13. Problem statement section is too weak. You should discuss the problem in more detail and explain the main superiority of the proposed model.
14. Manuscript is poorly written and contains many grammatical errors. It needs to be rewritten and proofread for grammatical errors by a native English speaker.
Minor Comments:
15. The texts written in Figure 1 are not clear and hence not readable.
16. The captions of the Figures 1 & 3 are too large, just like a paragraph. Please make it short and brief.
***
Author Response
Reviewer 2
We thank the reviewer for the recommendations provided. All provided suggestions were included in our manuscript.
Line # 51: For ‘that’ use ‘the aforementioned review’
We replaced ‘that’ with suggested wording.
Line # 63: use the word ‘in’ in the sentence
A professional English proofreading was performed.
Line # 68: delete the word ‘approaches’
This was a typing error and we removed the word ‘approaches’.
Line # 70: Use this statement in the manuscript ‘Present study had threefold objectives’
A professional English proofreading was performed.
Line # 132: Have some more references on grace period of two days
We included additional information on selection of choice grace period.
Lines: 142-148
A grace period designating when treatment could be initiated after baseline. Implementing a grace period makes the treatment strategies used in clinical practice more pragmatic [17,34]. The length of the grace period should rely on a clinical decision and serve to reduce heterogeneity between patients [35]. Our study defined the grace period during which patients may be eligible to begin treatment as two days after hospital admission. Consequently, patient’s ‘cloned’ data could appear in both treatment strategies within two days [36].
Line # 201: Kindly mention the package used in R software
Additional information on R packages is included.
Lines: 216-217
We used the survival, boot, and ggplot2 packages in R.
Kindly add a ‘limitation’ section also
We agree with the reviewer and created a separate Limitations section.
Lines: 411-421
- Limitations
Our study has important limitations. First, our main analysis relied on a straight-forward estimation of time-constant hazards enabling basic understanding of the data, the concept of a competing risks approach, and statistical quantities [32]. However, the assumption of time-constant hazards limited our knowledge of fluctuations in the treatment effect during the observational period. Second, time-varying confounding remained underestimated due to the lack of information on time-updated prognostic covariates. This is despite our attempt to reduce the potential of time-varying con-founding bias by choosing a grace period of two days and by considering as untreated those patients who initiated treatment later after the grace period. Furthermore, we assumed that our baseline confounders remained fixed within the grace period. Third, our simplified strategy was to compare initiators versus non-initiators, regardless of whether or not patients adhered to the treatment strategy to which they had been as-signed after the grace period. This implied the estimation of an intention-to-treat effect. Evaluating dynamic treatment exposures and per-protocol effect estimation requires information on adherence, data availability on time-updated covariates, and the application of complex analytical approaches for time-varying confounders, such as g-estimation-based methods [53,54]. Due to these shortcomings, we were unable to interpret treatment effects as per-protocol effect [35].
Reviewer 3 Report
The paper aims to emulate a target trial with a competing risk analysis using hospital-based observational data. The work is interesting but must be improved. As suggestions for improvement, I highlight the points below:
The work cites less than 31 articles, a low number for a high-impact article. Considering the relevance of the topic and the impact of this journal, authors should create a literature review section, presenting the state of the art on the subject in question and the main contributions and improvements of the proposed model.
In this sense, I suggest citing the newest risk analysis algorithms, statistical methods, and weighting techniques in the context of epidemiology, such as:
https://doi.org/10.3390/healthcare10112147
With this, I suggest that the authors highlight what the proposed methodology presents in terms of differences and improvements about those already existing in the literature.
The results and conclusions are very well structured and presented.
Author Response
Reviewer 3
The paper aims to emulate a target trial with a competing risk analysis using hospital-based observational data. The work is interesting but must be improved. As suggestions for improvement, I highlight the points below:
The work cites less than 31 articles, a low number for a high-impact article. Considering the relevance of the topic and the impact of this journal, authors should create a literature review section, presenting the state of the art on the subject in question and the main contributions and improvements of the proposed model.
We would like to thank the reviewer for their comments and suggestions. In line with the reviewer's feedback, we have included further details in the Introduction section regarding the three primary methodological approaches. Additionally, we have provided an in-depth explanation of the competing risk issue and the benefits of our extended approach.
In this sense, I suggest citing the newest risk analysis algorithms, statistical methods, and weighting techniques in the context of epidemiology, such as:
https://doi.org/10.3390/healthcare10112147
We would like to thank the reviewer for sharing with us a very interesting study. However, after careful consideration, we have decided not to include this study in our manuscript to avoid confusion with our inverse probability of censoring weights that was used to account for selection bias.
With this, I suggest that the authors highlight what the proposed methodology presents in terms of differences and improvements about those already existing in the literature.
We appreciate the reviewer's suggestions and revised the Discussion section accordingly. Specifically, we have included a comparison of our approach with other methods used in evaluations of treatment effectiveness used for patients with COVID-19.
Lines: 350-382
The importance of recognising competing events in both randomised trials and observational studies has been emphasized previously [8,9,45]. Our study highlights how essential it is to account for competing risks also in emulated target trials. In studies relying on routine clinical data, the follow-up of COVID-19 patients often ends with either in-hospital death or discharge alive. In our study, we assumed that hospital discharge would serve as a valid proxy for recovering from the disease. This assumption is based on hospital readmissions and post-discharge mortality rates in COVID-19 patients being relatively low [46]. Using a demonstrative data example, we estimated the treatment effects on in-hospital death by considering the effects on the competing event of hospital discharge rather than eliminating it. At the analysis stage, we provided cause-specific hazards and cumulative event probabilities according to previous recommendations [8,47]. We must admit that discharge might not present a methodological issue in COVID-19 studies in which survival data beyond hospital discharge is available. If such information was available in our data, our statistical methods could have been simplified because discharge would no longer have been a competing event for death. In that case for example, discharged patients remain in the risk set until a pre-specified time (e.g., date of last follow-up) [21].
Lin et al. compared three different analytical approaches to estimate survival probabilities in hospitalised patients with COVID-19 who received convalescent plasma therapy using observational data. They confirmed that the naïve (unweighted) Kaplan-Meier method yields biased results [48]. Similarly, in our naïve analysis, we showed that censoring patients at the time of the competing event leads to an overestimation of cumulative incidence probabilities for the primary outcome of both treat-ment groups. Therefore, applying standard survival analysis methods in the presence of competing risks is inappropriate [6,9,45,49]. Lin et al. also described two alternative approaches to account for competing events. Their first method assumed that the patients who were discharged would be alive until a pre-specified date. Their second method entailed an inverse probability weighted Kaplan-Meier estimator that account-ed for the eliminated (censored) competing event of discharge alive [48]. We believe that acknowledging the treatment effects on discharge alive is more meaningful: the interest may be to examine not only how treatment prevents death, but also how treatment effects the probability of recovering from the infection [41]. We showed that a competing risks analysis provides important insights on treatment effects on all clinically meaningful and heterogeneous endpoints [11,50].
The results and conclusions are very well structured and presented.
We thank the reviewer for the feedback.
Reviewer 4 Report
Martinuka et al. in their manuscript entitled "Target Trial Emulation Using Hospital-Based Observational Data: Demonstration and Application in COVID-19" show how three types of bias can be avoided in observational studies evaluating treatment effectiveness for patients hospitalized with COVID-19 via target trial emulation. The paper provides important and statistically valid information.
There are some points to be addressed:
Lines 18 - 27: I guess the Simple Summary as presented is part of the conclusion. I suggest to remove those lines and use the text in the conclusion.
Line 46: Introduction shall present all important references and relevant background. The authors must add and present the latest papers about target trial emulation and COVID-19. For instance, I added some possible references to be evaluated by the authors
Ioannou, G. N., Locke, E. R., O’Hare, A. M., Bohnert, A. S., Boyko, E. J., Hynes, D. M., & Berry, K. (2022). COVID-19 vaccination effectiveness against infection or death in a national US health care system: a target trial emulation study. Annals of Internal Medicine, 175(3), 352-361
Tran, V. T., Perrodeau, E., Saldanha, J., Pane, I., & Ravaud, P. (2021). Efficacy of COVID-19 vaccination on the symptoms of patients with long COVID: a target trial emulation using data from the ComPaRe e-cohort in France.
Breskin, A., Wiener, C., Adimora, A. A., Brown Jr, R. S., Landis, C., Reddy, K. R., ... & Brookhart, M. A. (2023). Effectiveness of Remdesivir Treatment Protocols Among Patients Hospitalized with COVID-19: A Target Trial Emulation. Epidemiology (Cambridge, Mass.).
Dickerman, B. A., Gerlovin, H., Madenci, A. L., Kurgansky, K. E., Ferolito, B. R., Figueroa Muñiz, M. J., ... & Hernán, M. A. (2022). Comparative effectiveness of BNT162b2 and mRNA-1273 vaccines in US veterans. New England Journal of Medicine, 386(2), 105-115.
Lines 387 - 394: Finally, the authors must improve the conclusion. It should demonstrate the significance of your findings. You may consider to use and extend the text from the Simple Summary to improve the conclusion.
Minor Comments:
Line 306: Figures S4, S5 and S6 may be added to the manuscript instead of being part of the supplementary material.
Line 321: Change text to Maringe et al. [16]
Author Response
Reviewer 4
Martinuka et al. in their manuscript entitled "Target Trial Emulation Using Hospital-Based Observational Data: Demonstration and Application in COVID-19" show how three types of bias can be avoided in observational studies evaluating treatment effectiveness for patients hospitalized with COVID-19 via target trial emulation. The paper provides important and statistically valid information.
We thank the reviewer for the feedback and for reviewing our manuscript.
There are some points to be addressed:
Lines 18 - 27: I guess the Simple Summary as presented is part of the conclusion. I suggest to remove those lines and use the text in the conclusion.
Following the suggestions of the reviewers, we removed the Simple Summary.
Line 46: Introduction shall present all important references and relevant background. The authors must add and present the latest papers about target trial emulation and COVID-19. For instance, I added some possible references to be evaluated by the authors
Ioannou, G. N., Locke, E. R., O’Hare, A. M., Bohnert, A. S., Boyko, E. J., Hynes, D. M., & Berry, K. (2022). COVID-19 vaccination effectiveness against infection or death in a national US health care system: a target trial emulation study. Annals of Internal Medicine, 175(3), 352-361
Tran, V. T., Perrodeau, E., Saldanha, J., Pane, I., & Ravaud, P. (2021). Efficacy of COVID-19 vaccination on the symptoms of patients with long COVID: a target trial emulation using data from the ComPaRe e-cohort in France.
Breskin, A., Wiener, C., Adimora, A. A., Brown Jr, R. S., Landis, C., Reddy, K. R., ... & Brookhart, M. A. (2023). Effectiveness of Remdesivir Treatment Protocols Among Patients Hospitalized with COVID-19: A Target Trial Emulation. Epidemiology (Cambridge, Mass.).
Dickerman, B. A., Gerlovin, H., Madenci, A. L., Kurgansky, K. E., Ferolito, B. R., Figueroa Muñiz, M. J., ... & Hernán, M. A. (2022). Comparative effectiveness of BNT162b2 and mRNA-1273 vaccines in US veterans. New England Journal of Medicine, 386(2), 105-115.
We thank the reviewer for providing the articles on the target trial emulation approach. We agree with the reviewer and included the available literature on the application of target trial emulation in COVID-19 research.
Lines: 71-74
During the COVID-19 pandemic, this framework was used in a wide range of evaluations, i.e., drug therapies [18–24], extracorporeal membrane oxygenation [25,26], vaccine effectiveness [27,28], non-pharmaceutical interventions, and policy evaluations [29]. To emulate a trial, there are two common methods taking the cloning approach or sequential trial approach [17,30].
Lines 387 - 394: Finally, the authors must improve the conclusion. It should demonstrate the significance of your findings. You may consider to use and extend the text from the Simple Summary to improve the conclusion.
Following the suggestions of the reviewer, we improved the Conclusions section.
Lines: 410-420
- Conclusions
We emulated this trial via an extended competing risk analysis applicable for evaluating the effectiveness of a treatment for hospitalised patients with COVID-19, by relying on routine register-based observational data. Our findings demonstrate that this methodology can overcome three major methodological challenges: immortal-time bias, confounding bias, and competing risks. Our approach enables potential treatment effects to be evaluated on all clinically relevant outcomes and heterogeneous endpoints. We recommend avoiding application of the naïve Kaplan-Meier survival analysis method in the presence of competing events, as that can lead to exaggerated estimates. This paper and its supporting materials provide technical guidance for trial emulations involving competing risks analysis.
Minor Comments:
Line 306: Figures S4, S5 and S6 may be added to the manuscript instead of being part of the supplementary material.
We thank the reviewer for the suggestions provided. After careful consideration, we decided not to include the additional results shown in Figures S4, S5, and S6 to avoid any misinterpretation. We want to avoid mixing the two approaches used in our study. Additional analysis was performed to show the available statistical methods and to compare the results with those of the main analysis.
Line 321: Change text to Maringe et al. [16]
The text is replaced with a short form.
Line: 344
In our study, the trial emulation with a competing risks model is proposed and ap-plied to a setting where hospital data is available by adapting the step-by-step tutorial presented by Maringe et al. [31].
Round 2
Reviewer 1 Report
No comment
Reviewer 3 Report
I suggest accepting the paper for publication.